# 16S rRNA Gene Amplicon Sequencing Data of the Iron Quadrangle Ferruginous Caves (Brazil) Shows the Importance of Conserving This Singular and Threatened Geosystem



Camila G. C. Lemes [1], Morghana M. Villa [2], Érica B. Felestrino [1], Luiza O. Perucci [3], Renata A. B. Assis [1], Isabella F. Cordeiro [1], Natasha P. Fonseca [1], Lara C. C. Guerra [2], Washington L. Caneschi [1], Lauro Â. G. Moraes [3], Flávio F. do Carmo [4], Luciana H. Y. Kamino [4], Pedro N. C. Vale [5], Suzana E. S. Guima [6], João C. Setubal [6], André A. R. Salgado [5] and Leandro M. Moreira [1,2,*]

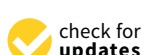

1    Programa de Pós-Graduação em Biotecnologia, Núcleo de Pesquisas em Ciências Biológicas, Universidade Federal de Ouro Preto, Ouro Preto 35400-000, MG, Brazil; camilalemes.mail@gmail.com (C.G.C.L.); erica_felestrino@yahoo.com.br (É.B.F.); renatab.assis@uol.com.br (R.A.B.A.); isabella.fercod@gmail.com (I.F.C.); natashapfonseca@gmail.com (N.P.F.); washingtoncaneschi@gmail.com (W.L.C.)

2    Departamento de Ciências Biológicas, Instituto de Ciências Exatas e Biológicas, Universidade Federal de Ouro Preto, Ouro Preto 35400-000, MG, Brazil; morghana_villa_@hotmail.com (M.M.V.); laraguerra.bio@gmail.com (L.C.C.G.);

3    Laboratórios Multiusuários de Genômica e Bioinformática, Universidade Federal de Ouro Preto, Ouro Preto 35400-000, MG, Brazil; luizaperucci@gmail.com (L.O.P.); lauro_moraes@hotmail.com (L.Â.G.M.)

4    Instituto Prístino, Belo Horizonte 30642-180, MG, Brazil; carmo.flaviof@gmail.com (F.F.d.C.); lucianakamino@gmail.com (L.H.Y.K.)

5    Departamento de Geografia, Instituto de Geociências, Universidade Federal de Minas Gerais, Belo Horizonte 31270-901, MG, Brazil; pncvgeo@yahoo.com.br (P.N.C.V.); aarsalgadoufmg@gmail.com (A.A.R.S.)

6    Departamento de Bioquímica, Instituto de Química, Universidade de São Paulo, São Paulo 05508-000, SP, Brazil; suzy.eiko@gmail.com (S.E.S.G.); setubal@iq.usp.br (J.C.S.)

*    Correspondence: lmmorei@gmail.com; Tel.: +55-31996312710

**Abstract:** The Iron Quadrangle (IQ) is one of the main iron ore producing regions of the world. The exploitation of its reserves jeopardizes the high biological endemism associated with this region. This work aimed to understand the diversity and bacterial potential associated with IQ caves. Floor and ceiling samples of seven ferruginous caves and one quartzite cave were collected, and their microbial relative abundance and diversity were established by 16S rRNA gene amplicon sequencing data. The results showed that ferruginous caves present higher microbial abundance and greater microbial diversity compared to the quartzite cave. Many species belonging to genera found in these caves, such as *Pseudonocardia* and *Streptacidiphilus,* are known to produce biomolecules of biotechnological interest as macrolides and polyketides. Moreover, comparative analysis of microbial diversity and metabolic potential in a biofilm in pendant microfeature revealed that the microbiota associated with this structure is more similar to the floor rather than ceiling samples, with the presence of genera that may participate in the genesis of these cavities, for instance, *Ferrovum*, *Geobacter*, and *Sideroxydans*. These results provide the first glimpse of the microbial life in these environments and emphasize the need of conservation programs for these areas, which are under intense anthropogenic exploration.

**Keywords:** ferruginous outcrops; natural cavities; canga; microbiome; microorganisms



## 1. Introduction

For many researchers, karst is a type of landscape that only occurs in carbonates. However, during the last decades several karst landscapes in non-carbonate rocks have been identified in South America [1–5]. Although most of these landscapes are located on siliciclastic rocks, karstic geoforms, mainly caves, have also been found in iron ore, such as

banded iron formation and canga (superficial weathering product). Many of these iron ore caves are in the Iron Quadrangle (IQ) in southeastern Brazil [6]. In IQ, although most caves are small, they are high in density and several have strong evidence that the dissolution processes were important to their morphogenesis [7–10]. This allows to classify the region as karstic or, at least, pseudo-karstic.

The IQ is located in the center-south region of the State of Minas Gerais, Brazil, and corresponds to an area of approximately 7200 km$^2$ [11] (Figure 1) that share two biodiversity hotspots: the Cerrado (Brazilian Savanna) and the Atlantic Forest [12] (Supplementary Figure S1). The ironstone ranges, which are geologically formed by banded iron formation (BIF) of itabirites with very variable iron concentration of Archean and Paleoproterozoic ages, occur in this region. The iron concentration in itabirites is variable as this rock has two bands: one composed by hematite ($Fe_2O_3$) and other by quartz ($SiO_2$) and the thickness of each one of these bands naturally varies. Duricrusts, known as cangas, formed by minerals resulting from the natural alteration of itabirite, such as limonite and goethite ($FeO(OH)$) [10,13], cover the upper reaches of ironstone ranges and reach up to 30 m in thickness [13,14].

The ironstone ranges are subject to abiotic stressors such as acid substrates and possess anomalous metal contents. These abiotic stressors are due to the natural weathering process of BIF's that transforms the hematite, present in the itabirite, into limonitic capping and in laterized clay deposits, which are rich in goethite. The associated soils are extremely oligotrophic and depleted of exchangeable nutrients, such as calcium, potassium, magnesium, and total nitrogen, resulting in an environment with intense ecological filters, and harboring a high number of endemic species that are rare and endangered [15–17].

The IQ also shelters hundreds of ferruginous caves and is one of the regions with the highest occurrence of these geoforms in the word. In general, although they take a long time to form, ferruginous caves are small in size, as they rarely exceed 100 m in horizontal projection, with an average of 61 m$^2$ of area and 81.3 m$^3$ of volume and 3 m of unevenness in the cave elevation along its horizontal projection [18].

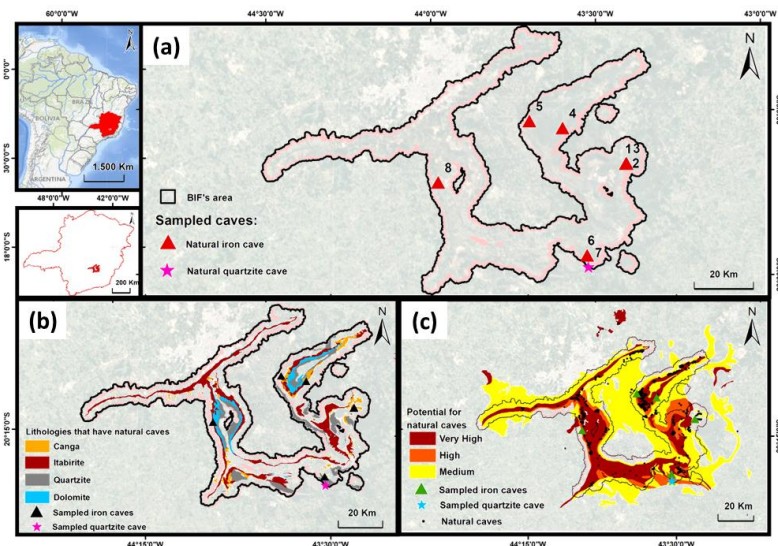

**Figure 1.** Thematic maps of the IQ region, MG, Brazil. (**a**) IQ delimitation and location of the eight caves investigated in this study (Details see Table 1); (**b**) Lithologies that have natural cavities; (**c**) Potential for the formation of natural cavities. All maps were obtained from "Instituto Prístino: atlas digital geoambiental. WebGis system (Web Geographical Information System) of free access to the environmental database". Available online: https://institutopristino.org.br/atlas accessed on 15 June 2020.

**Table 1.** Characterization of sampled caves.

| Cave Number | Localization | | | | | | Geomorphology | | | Other Features | | |
|---|---|---|---|---|---|---|---|---|---|---|---|---|
| | County | Locality | Biome | Latitude (S) | Longitude (W) | Altitude (m) | Lithology | Ceiling | Floor | Human Activity | Proximity to Mine (up to 500 m) | Protected Area |
| 1 | Mariana | Chapada | Atlantic Forest | 20°9′53.78″ | 43°24′19.49″ | 879 | Ferruginous | Canga | Canga | No | ~250 m | No |
| 2 | Mariana | Chapada | Atlantic Forest | 20°9′54.03″ | 43°24′29.72″ | 865 | Ferruginous | Canga | Canga | No | ~350 m | No |
| 3 | Mariana | Chapada | Atlantic Forest | 20°9′49.15″ | 43°24′21.85″ | 880 | Ferruginous | Canga | Canga | No | ~350 m | No |
| 4 | Caeté | Gandarela's Mountain | Atlantic Forest | 20°3′19.58″ | 43°41′42.41″ | 1624 | Ferruginous | Canga | Itabirite | No | No | Yes |
| 5 | Santa Bárbara | Gandarela's Mountain | Atlantic Forest | 20°3′23.40″ | 43°35′59.94″ | 1236 | Ferruginous | Canga | Itabirite | No | No | Yes |
| 6 | Ouro Preto | Lavras Novas | Atlantic Forest | 20°26′35.76″ | 43°31′31.18″ | 1480 | Ferruginous | Canga | Itabirite | No | No | No |
| 7 | Ouro Preto | Lavras Novas | Atlantic Forest | 20°28′42.20″ | 43°31′15.72″ | 1346 | Quartizite | Quartizite | Quartizite | No | No | No |
| 8 | Nova Lima | Moeda's Mountain | Atlantic Forest | 20°13′18.26″ | 43°58′38.48″ | 1488 | Ferruginous | Canga | Itabirite | Yes | No | No |

The small diameter entrances of these caves are almost always located at the edges of the canga as a result of erosive processes or, less frequently, are small vertical openings caused by collapse of the canga mantle [7]. Although small, they have a very diversified geological composition, found only in BIFs, in cangas, or in contact with both; they generally have a ceiling composed by the canga and a floor formed by the BIF [10]. Despite their distinct composition, their floors are usually filled with ferruginous sediments (rich in hematite, limonite, and goethite) of granulometries ranging from clay to boulders [18]. Its speleogenesis may be related with many processes [19], but the most recent paper defends a hypogenic genesis related to late vadose stages where the erosion only exhumes the cave [20]. Most of the IQ caves, despite not having an evident entrance, are not completely isolated from the surface. As the canga is very porous and the itabirite is fractured, flows of superficial waters enable penetration of micro particles inside the caves. The interiors of the caves present considerable aphotic extensions, and most of these have stable environmental conditions such as temperature, pH, and humidity, providing a highly selective environment for life [21]. Although the biofilm in pendant microfeature in ferruginous cavities may be rare [22,23], these structures are defined as secondary mineral deposits originating from the processes of circulation and discharge of water or from biological activity [24].

Despite its biological and speleological importance, the IQ is one of the natural areas in Brazil most endangered with a loss and degradation of natural ecosystems, due to the intense mining activity and urban expansion. The region concentrates one of the world's largest open pit complexes [25,26] (Supplementary Figure S1). In BIF areas, there is a defined relationship between the occurrence of iron ore and high-grade ore caves, as a result of which, the full exploitation of iron resources will lead to the loss of caves [27]. The industrial waste generated during mineral processing has several impacts: pollution of groundwater, rivers, and soil by metals and trace elements (e.g., Cr, Fe, Mn, and Zn); alterations to water turbidity; as well as silting of stretches of streams located downstream of the dams. In addition, thousands of tons of chemicals used for the treatment and processing of minerals (such as caustic soda, sodium hypochlorite, ether, and hydrogen peroxide) are capable of altering the natural conditions of water bodies, soil, biota, ecosystems, and human health [28]. Additionally, in the last years, discharges of tailing dams have placed the IQ, causing disasters as well as irreparable environmental damage [29,30]. However, there are still no studies that reveal the environmental impacts and damages caused by scale mining industry. Therefore, it is likely that such influences have been source of losses of microbial biodiversity and cave invertebrates.

Studies reporting the presence and importance of the microbiota in ferruginous caves are still incipient. Parker and colleagues have made significant contributions to the importance of the microbiota in the biogenesis of ferruginous cavities [23,31]. According to these authors, the action of these bacteria when reducing iron III to iron II allows their solubilization and subsequent leaching through groundwater, which, taking advantage of the system of rock failures and fractures and the microporosities of the canga, penetrates and seeps into the interior the cave. Thus, expanding knowledge about the diversity and importance of the microbiota associated with these cavities, especially in the IQ region, provide important characterization of the ecosystems within these unique environments [32] (Supplementary Figure S1).

In this context, this work aimed to investigate the prokaryotic diversity associated with the natural subterranean cavities in ferruginous rocks by 16S rRNA gene amplicon sequencing, and to predict the characteristics and metabolic potentials of the microorganisms identified in this environment. The reported data will contribute to a broader initiative that aims to establish conservation areas within the IQ in the future. In addition, the data may support the conservation in situ of iron caves into newly protect areas or to stimulate studies in few caves located in conservation areas, such as the Gandarela National Park, the main preservation area in the entire region, and the Natural Monument of Serra da Moeda [33].

## 2. Materials and Methods

### 2.1. Collection Sites

The determining factors in the selection of caves included geographic location, variation in altitude, and degree of human interference (Table 1 and Figure 1). Selection of the quartzitic cave used as an experimental control took into account the geographic proximity and lithology similarity to the IQ. According to the literature, caves are stable environments, where changes in temperature and humidity are lesser than those in the atmosphere, and therefore the internal temperature range between 11 °C and 22 °C and humidity between 48% and 98% in the IQ ferruginous caves throughout the year, while pH ranges from 3.5 to 4.9 [34]. The samples were collected at points close to the cavity entrances, therefore in a borderline region between the photic and aphotic zones (when possible), and in different aphotic regions depending on the dimensions of the cavities.

Samples of 50 g of solid material from the floor, ceiling, and biofilm hanging from the ceiling were collected from seven caves associated with canga outcrops and a quartzite cave (experimental control), using previously sterilized materials such as spatulas and 50-mL Falcon® tubes (Life Sciences, Corning, NY, USA).

### 2.2. Total Genomic DNA Extraction

Total microbial DNA was extracted from 0.25 g of each sample using the PowerSoil DNA Mobio™ kit (GeneWorks, Adelaide, South Australia, Australia), following the protocol recommended by the manufacturer. After extraction, the DNA samples were quantified and their purity was analyzed using NanoDrop™ (Thermo Fisher Scientific, Waltham, MA, USA). The sample quality was verified by electrophoresis on 0.8% agarose gel. No blank samples were used as a contamination control associated with the sequencing kit reagents [35].

### 2.3. Partial Amplification of the 16S Ribosomal Gene

To amplify 16S ribosomal genes of the V4-V5 region, 515F and 926R oligonucleotides were used [36]. These oligonucleotides were designed to include the adapter sequences used in the Ion Torrent sequencing protocol and were linked to a 10 nt barcode for each sample [36]. The complete list of oligonucleotides used in this study can be found in Supplementary Table S1. The PCR was prepared in a final volume of 30 μL containing 0.38 mM dNTP, 2.5 mM $MgCl_2$, and 0.15 μM of each oligonucleotide, 20 ng DNA, 0.08 U of Platinum™ Taq DNA polymerase (Invitrogen, Waltham, MA, USA), and 10× Buffer. Amplification was performed in a thermocycler Biocycler® (Applied Biosystems, Waltham, MA, USA) with an initial cycle of 5 min at 94 °C, followed by 35 cycles of 30 s at 94 °C, 1 min at 57 °C, and 1 min at 72 °C, with a final cycle of 5 min at 72 °C. The amplicons were analyzed on an 0.8% agarose gel. Once quality was checked, 22 μL of the final PCR product was purified using the GFX PCR Kit™ and Gel Band Purification™ (GE Healthcare, Chicago, IL, USA) Kit, following manufacturer's recommendations. The quality of the purified product was again measured on 0.8% agarose gel.

### 2.4. 16S rRNA Amplicon Sequencing Using the Ion Torrent Platform

The DNA library was quantified using Qubit dsDNA HS™ (High Sensitivity) using the Qubit 2.0™ fluorometer (Life technologies, Corning, NY, USA) and the samples were then normalized to 70 pM. For library assembly, 20 μL of each sample was placed in a single 1.5 mL tube and homogenized. Then, 25 μL of this DNA pool was loaded onto a 318 v2™ chip (Life technologies, Corning, NY, USA), and the samples were sequenced using the Ion PGM™ platform (Life technologies, Corning, NY, USA) following the manufacturer's recommendations. After sequencing, the polyclonal sequences were filtered using the software Torrent Suite™, version 5.0.5, and the data obtained were exported as FastQ files. The homopolymers, stretches of the same nucleotide sequence, are also detected at very high accuracy by the Ion technology [37]. A 5-mer is currently called with greater 97.5%

per base accuracy according to the IonTorrent Platform, although several studies report that there should be a concern with the theme [37–40].

### 2.5. 16S rRNA Amplicon Sequencing Data Analyses

The data were processed using the clustering-based pipeline adapted from the Brazilian Microbiome Project (BMP) [41]. In the pre-processing step, reads were filtered using Prinseq [42]. Low quality reads of less than 100 bp, and with an average quality (Q) $\leq$ 20 were discarded. Filtered reads were dereplicated using the USEARCH v.8.1 script [43], performing clustering of abundant sequences and discarding singletons (single unique reads, without similarity). For rarefaction analysis, we adopted the sequencing depth of 28,252 reads for all samples, which was the number of reads of the smallest sample. After rarefaction, the reads were grouped into OTUs (Operational Taxonomic Units) following the UPARSE method [44], where the identity of sequences corresponding to the same OUT was $\geq$97%. For each OTU generated, a taxonomy was assigned using the RDP 11.5 (Ribosomal Database Project) classifier [45] available in QIIME 2 v.2021.2 [46]. Additionally, all sequences associated with chloroplasts were manually excluded from the analyses. A table was then obtained in the BIOM (The Biological Observation Matrix) format containing the corresponding read counts for the OTUs.

Alignment of OTUs was processed by PyNAST [47], available in QIIME 2. A phylogenetic tree was constructed using FastTree [48], also available in QIIME 2. Diversity analyses were performed from the BIOM file and the phylogenetic tree to generate the results of alpha-diversity, beta-diversity, frequency tables, and taxonomic classification charts. For the rarefaction plot we used the alpha pipeline from the diversity plugin available on QIIME 2 v.2021.2 [46]. To obtain the observed features for all samples, the following groups were compared: cave vs. cave (regardless of coming from ceiling or floor), ceiling vs. floor (regardless of the cave), and ceiling vs. floor separated by caves. The Kruskal–Wallis test [49] was used for these comparisons (alpha-group-significance visualizer from the diversity plugin).

The Phyloseq package v.1.34.0 [50] was used for alpha and beta diversity analysis. From the BIOM table imported into R software, alpha diversity metrics such as observed features, Shannon [51], Simpson [52], and Chao1 [53] were calculated and plotted using the function plot_richness from phyloseq. For ordination plots of beta diversity metrics, sampling counts from the imported BIOM were first transformed to even sampling depth with transform_sample_counts function (formula: $1 \times 10^6 \cdot x/\mathrm{sum}(x)$, where x is the number of OTU counts in a sample, and sum(x) is the sum of all OTUs counts in a sample). Ordinate function from phyloseq was then used to calculate both unweighted and weighted Unifrac distance [54] and the graphs of the Principal Coordinate Analysis (PCoA) were generated from these distances [55]. Ellipses were computed for the ordination plot with stat_ellipse function from ggplot2 v.3.3.3 [56] considering a multivariate t-distribution with 0.95 level. For beta diversity statistical analysis, we used the diversity plugin from QIIME2. Unweighted Unifrac and weighted Unifrac [54,55] distances were calculated between samples and added into distance matrices (beta and beta-phylogenetic pipeline). ANOSIM and PERMANOVA tests [57] were used with 999 permutations in order to compare the ceiling group vs. the floor group and one cave vs. another cave (beta-group-significance visualizer with pairwise option from the diversity plugin). Biofilm in pendant microfeature sample was not included for these tests, as the material associated with the biofilm structure only allowed for one sample to be collected.

All Metagenome sequences were deposited in the Sequence Read Archive under Bioproject PRJNA575049. The biosample IDs (SAMN12876970 to SAMN12876998) and their corresponding cave names are present in the Supplementary Table S1.

### 2.6. Functional Metabolism Prediction from 16S rRNA Amplicon Sequences

To perform functional metabolic predictions, we used the program PICRUSt2 [58]. A Nearest Sequenced Taxon Index (weighted NSTI) of score 0.03 was used as default.

Functional classification was based on KEGG [59], which classifies the metabolic pathways into three possible levels, as detailed below. The analysis was performed for Level 1 categories: Metabolism, Cell Processes, Cell Process/Signaling, Environmental Information Processing, and their respective subcategories (levels 2 and 3). Statistical analyses were performed using STAMP (STatistical Analysis of Metagenomic Profiles) software 2.1.3 [60], for multiple groups ANOVA was used followed by post hoc Tukey–Kramer test ($p < 0.05$) and for two groups Welch's *t*-test was used ($p < 0.05$).

## 3. Results

### 3.1. Sample Quality and the Number of Sequences Obtained

Thirty samples were collected from seven ferruginous and one quartzite caves: 16 samples from the floor, 13 from the ceiling, and one from a biofilm in pendant microfeature (Table 1 and Figure 1). Sample 18 was disregarded from the analysis for technical problems with hybridization of the specific primer-barcode. No blank samples were used as a contamination control. More than 8 million reads were generated, with fragments varying between 100 and 450 bp (featuring a bimodal distribution in the lengths of 200 bp and another in 400 bp), totaling 76% of the total capacity utilization of the chip (Supplementary Table S1). After removal of the smaller-size sequences (between 100 and 300 bp, thus eliminating the reads that made up the first peak of the read distribution), 6 million reads were considered valid (74% of the chip capacity), distributed among 29 samples with coverage varying between 84,000 and 358,000 reads per sample (Supplementary Table S1).

### 3.2. Alpha Diversity Characterization

Rarefaction curves were determined for each cave based on Observed_OTUs (Figure 2a). Cave 1, followed by caves 5, 6, and 4, were those that presented the highest number of observed OTUs, whereas cave 7 (quartzite), together with cave 8 and 3, presented the fewest. The three representative caves of Chapada de Canga (1, 2, and 3) presented distinct rarefaction curves. Despite the observed differences, there was no statistical difference between any of the rarefaction curves.

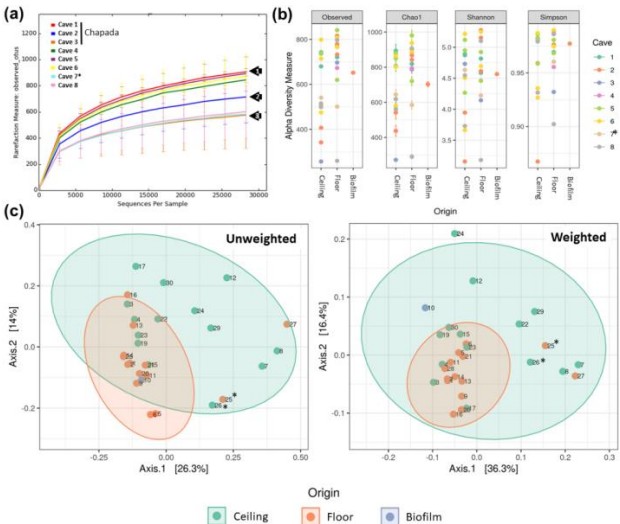

**Figure 2.** Global analysis of the 16S rRNA amplicon sequencing profile of the samples. (**a**) Rarefaction curve using the Observed_Otus metric ($p < 0.05$). (**b**) Diversity indices for the characterization of the prokaryotic community associated to cave samples. The expanded version of these results is shown in the Supplementary Figure S2. (**c**) Analysis of beta-diversity by PcoA using unweighted and weighted unifrac distance ($p < 0.05$). Orange line highlights the grouping of floor samples and green line the ceiling samples. Quartzitic cave samples are separated from ferruginous cave samples. Note that the floor sample from cave 8 (sample 27) is distant from all other samples (see details at Supplementary Figure S3). * quartzite cave samples.

The various measures of alpha diversity for a given cave yielded similar results (Figure 2b and Supplementary Figure S2A,B). General observations that can be made are that floor samples tend to be more diverse than ceiling samples, although there are no statistical differences; and that there was substantial variation in alpha diversity in samples from the same cave, especially for caves 2 and 8.

### 3.3. Beta-Diversity Characterization

PCoA (unweighted and weighted Unifrac, Figure 2c) analysis of samples showed that floor samples could be clustered together more tightly than ceiling samples, with statistical difference between ceiling and floor (Supplementary Table S2). Outliers in the case of floor samples were 25 (quartzite cave) and 27 (cave 8). Note also that its ceiling sample (26) was placed next to its floor sample, indicating that these samples have similar composition. This was not the case for cave 8, as its ceiling samples (29 and 30) were placed far from the floor sample.

### 3.4. Comparison of Prokaryotic Community Structures among Caves

The composition of the communities in each cave was investigated. It was found that between 14 and 26 phyla associated with 101 to 183 genera can be found per investigated sample, with a number of non-redundant phyla varying between 23 and 27, and non-redundant genera ranging from 161 to 216 per investigated cave (Supplementary Tables S3 and S4). In the ferruginous caves, the most abundant phyla were Acidobacteria (43.6%), Chloroflexi (15.4%), Proteobacteria (14.8%), Actinobacteria (4%), Nitrospirae (2.0%), and Firmicutes (1.8%), which together represented 86% of the entire prokaryotic community (Figure 3). The most representative classes in the ferruginous caves were Alphaproteobacteria (16.4%), Betaproteobacteria (15.8%), and Actinobacteria (15.4%), whereas Gammaproteobacteria (16%), Betaproteobacteria (15%), and Actinobacteria (15%) dominated in the quartzite cave (Figure 4a). The genera with greatest relative abundance in the quartzite cave were *Aciditerrimonas* and *Leucobacter* (Actinobacteria), *Bryobacter,* and GP6 (Acidobacteria), *Alcaligenes* (Betaproteobacteria), *Jahnella* (Deltaproteobacteria), *Coxiella*, *Rhodonobacter,* and *Xanthomonas* (Gammaproteobacteria), *Gemmatimonas* (Gemmatimonadetes), and *Nitrospira* (Nitrospira). In contrast, the most abundant genera in the ferruginous caves were *Blastopirellula* (Planctomycetia), WPS1 (WPS), and *Beijerinckia* (Alphaproteobacteria) (Supplementary Tables S3 and S4 and Figure 4b).

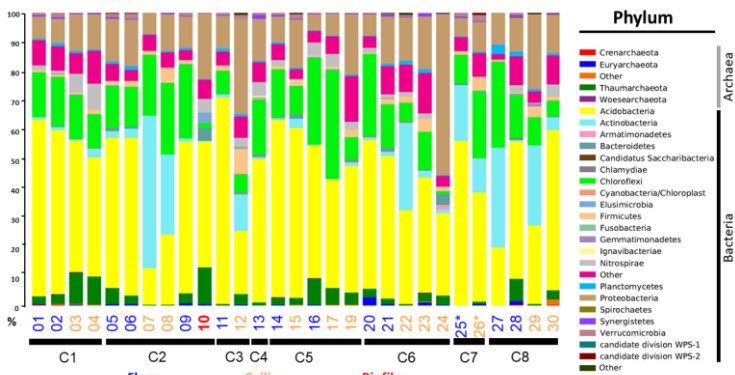

**Figure 3.** Diversity of the phyla present in the ferruginous and quartzitic caves, obtained from the table of OTUs. Acidobacteria, Chloroflexi, Proteobacteria, Actinobacteria, Nitrospirae, and Firmicutes showed relative abundance > 1% in the composition of the total samples.

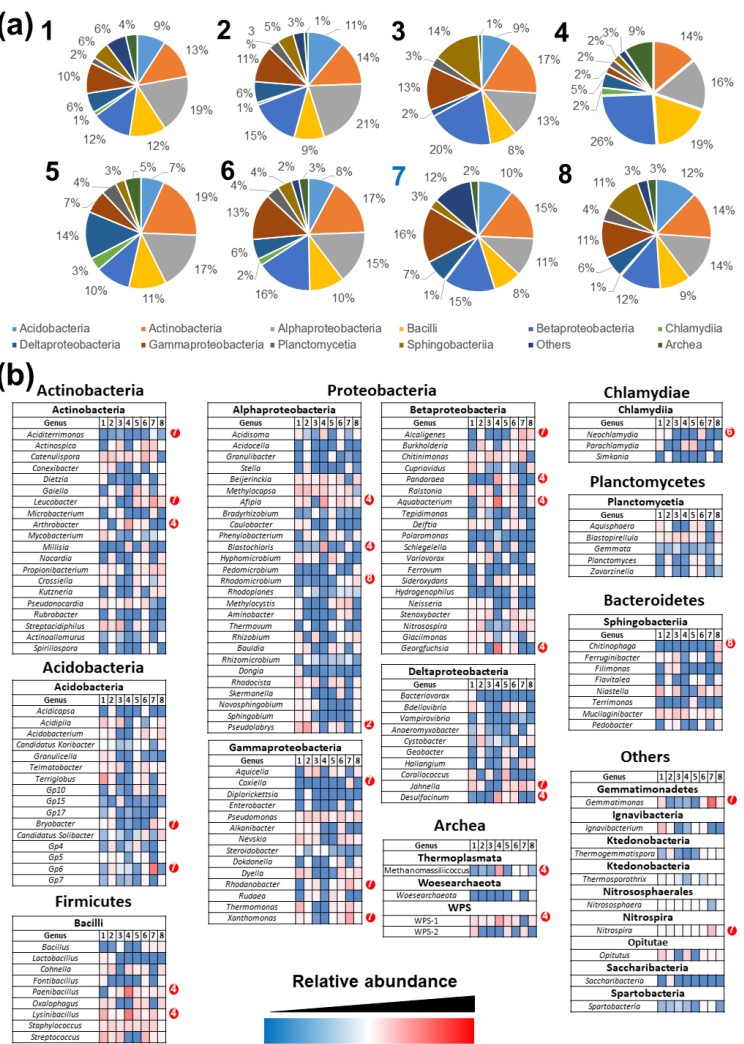

**Figure 4.** Relative abundance and diversity analysis. (**a**) Relative abundances and bacterial diversity at class level in ferruginous and quartzitic caves. The numbers 1 to 8 indicate the eight caves investigated in this study and georeferenced in Figure 1. (**b**) Heatmap of genus abundances (trend) in the samples collected in caves 1 to 8 (according to Table 1). The numbers in red circles indicate which cave presented greater relative abundance of the genus in comparison to other investigated caves.

## 3.5. Metabolic Analysis

A general metabolic prediction analysis identified similarities in metabolic potential between the ferruginous microbiota and the quartzite cave microbiota (Figure 5A). However, some categories of biotechnological interest, when analyzed separately showed statistical differences between the quartzite cave and ferruginous cave samples (Figure 5B). Within all 61 metabolic pathways associated with the level Metabolism (level 1), four subcategories on level 3 are highlighted: novobiocin biosynthesis (Biosynthesis and Biodegradation of secondary metabolites), nitrogen metabolism (Energy metabolism), biosynthesis of 12, 14, and 16 membered macrolides, and biosynthesis of type II polyketide products (Metabolism of Terpenoids and Polyketides). Regarding categories associated with (i) Cellular Processes/Signaling and (ii) Environmental Information and Processing (all level 1), we highlight the level 3 subcategories involved in Signal Transduction Mechanisms (Signal Transduction) and Secretion System (Membrane Transport) (Figure 5B). The samples from the ferruginous caves showed a higher proportion of sequences for the categories signal transduction mechanisms, nitrogen metabolism, novobiocin biosynthesis, and secretion system. For the pathways: biosynthesis of 12, 14, and 16 membered macrolides

and biosynthesis of type II polyketide products the highest proportions were from the quartzite cave.

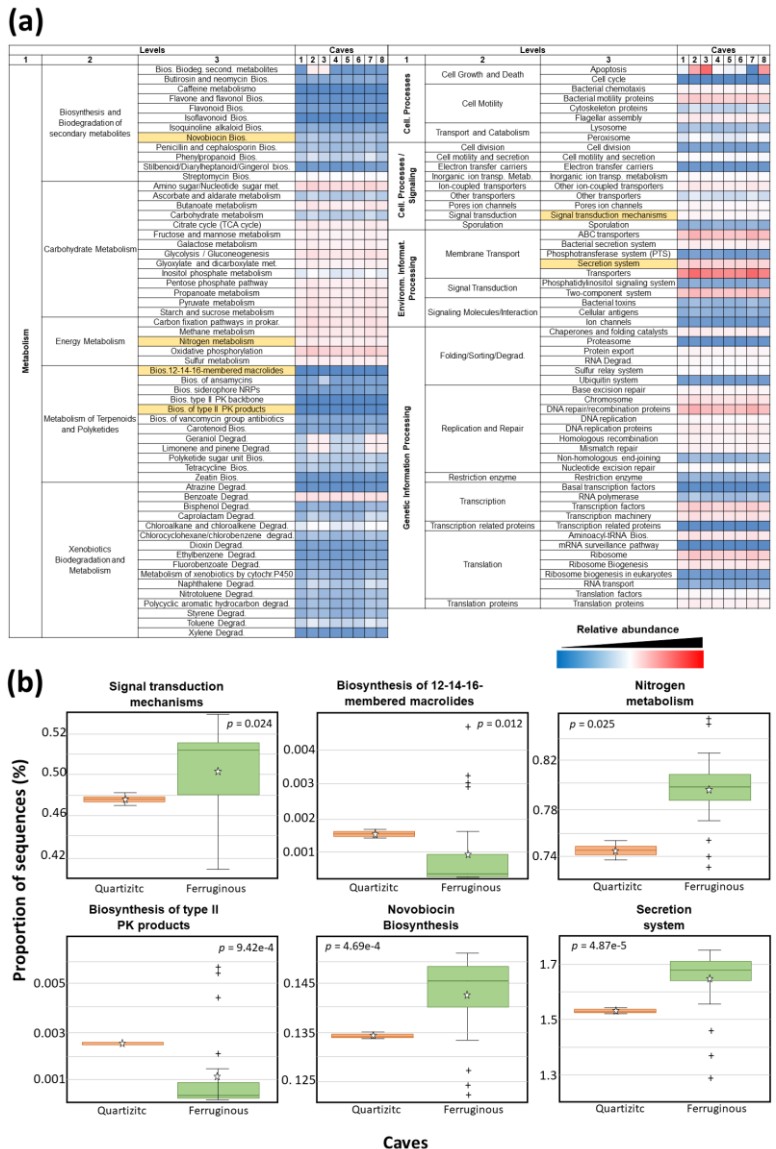

**Figure 5.** Relative proportions of the functional metabolic profiles based on sequences for ferruginous and quartzite caves. (**a**) Relative abundance of sequences between the caves based on Picrust predictions for functional categories inferred at level 3. Categories highlighted in yellow are presented in detail in b. (**b**) Comparative analysis of proportion of sequences (%) for predicted metabolic functions on genus level between the ferruginous caves and the quartzite cave. Welch's *t*-test was used ($p < 0.05$).

### 3.6. Ferruginous Caves Associated with Chapada de Canga

Three of the eight caves investigated in this study are located in the Chapada de Canga region (Figure 6a). Despite their geographical proximity, the observed microbiological richness differed between these three caves. Although no statistical difference was observed while cave 1 presented highest richness (Figure 2a), cave 3 presented the least richness and still showed the highest diversity among the caves in this region (with 32 specific OTUs) (Figure 6b). Considering only the samples of the cave floor, the highest diversity (at the genus level) was observed in cave 2, with 27 specific OTUs (2.1% of all OTUs) (Figure 6c). Specific OTUs are those identified only at that location when compared to the other Chapada de Canga caves in this study. The samples from cave 3, with 38 specific

OTUs (22.5% of all OTUs) (Figure 6d), showed the highest diversity among the ceiling samples. Among the specific OTUs of the ceiling of each cave, cave 2 stands out due to its high relative abundance of Actinobacteria (30%). Less abundances of Actinobacteria, 0 and 19%, were observed in the ceiling samples of cave 1 and 3, respectively.

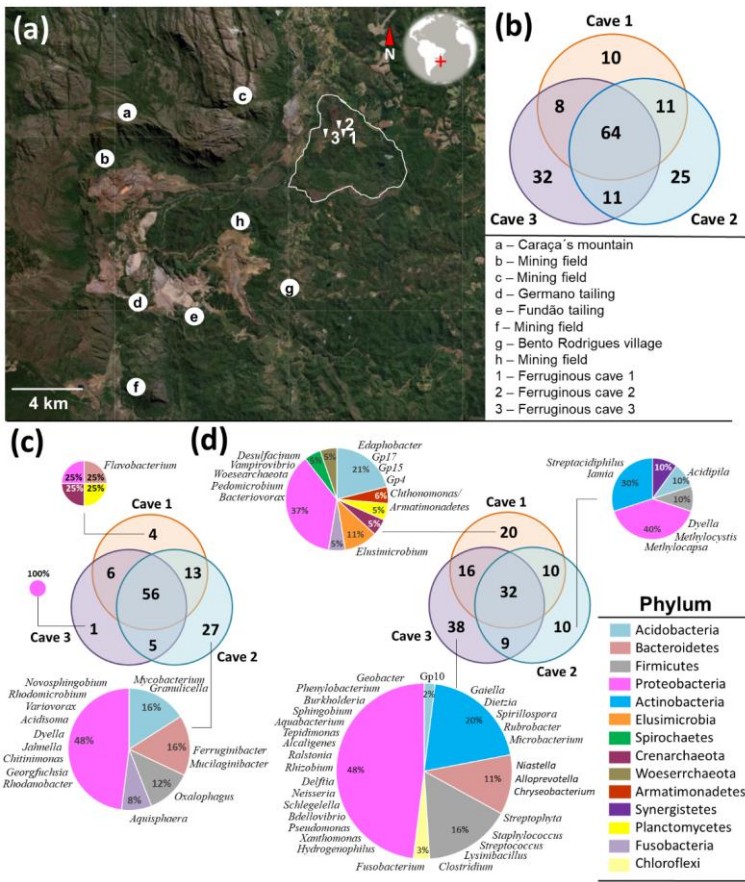

**Figure 6.** Geographical location of caves of Chapada de Canga and relations between genera determined in the caves. (**a**) A site map of Chapada de Canga caves. For more details see Figure 1. The map was obtained from "Instituto Prístino: atlas digital geoambiental. WebGis system (Web Geographical Information System): https://institutopristino.org.br/atlas, accessed on 15 June 2020. (**b**) Venn diagram representing the number of OTUs (on genus level) shared by the three caves of Chapada de Canga. (**c,d**) Venn diagrams representing unique and shared genera between each Chapada cave for floor and ceiling, respectively.

Cave 2 was analyzed in detail as it is the only one to present a biofilm hanging from the ceiling. When comparing the floor, ceiling, and biofilm samples obtained in cave 2, it was possible to identify different number of specific OTUs. A greater number of specific genera (24) was observed in the floor samples, compared with 10 and 9 in the biofilm and ceiling, respectively (Figure 7a). Interestingly, the number of genera shared exclusively between the floor and biofilm (26) and between the floor and ceiling (25) was much larger compared to the number of genera shared between the ceiling and biofilm (1), which are physically connected. In order to make this correlation more evident between the microbiota of the floor and biofilm, functional prediction of microbiota metabolism involving the three substrates was analyzed. Among the 24 categories investigated, 21 presented direct correspondence between the floor and biofilm, with emphasis on the pathways of secondary compounds and cellular processes (Figure 7b).

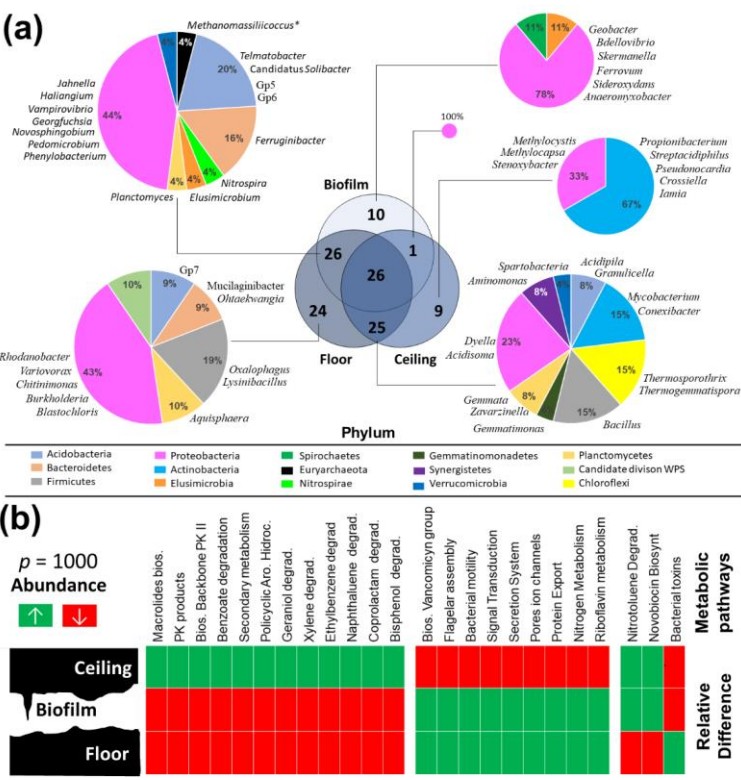

**Figure 7.** Diversity and prediction of functional metabolism in cave 2. (**a**) Venn diagram showing relations between floor, ceiling, and biofilm hanging from the ceiling samples from cave 2 (Chapada de Canga), on phylum level. (**b**) Analysis of the functional prediction and metabolic pathways associated with the ceiling, floor, and biofilm of cave 2. Elevated abundances for each pathway are shown in green, reduced abundances in red. ANOVA followed by post hoc Tukey–Kramer test ($p < 0.05$), however no statistical difference was observed to the categories available.

### 3.7. Inference of Ecological Relations in Ferruginous Caves

A systematic analysis of metabolic capacity of each genus identified in cave 2 (Supplementary Table S5) was performed to establish a model that indicates the importance of the microbiota for maintaining ecological relations within the ferruginous caves.

## 4. Discussion

### Ferruginous Caves in IQ

The physical environmental heterogeneity inside the iron cavities is directly reflected in their microbiological biodiversity, and provides unique habitats with pronounced differences in resources, with some being wet and others with and without light [23,31]. In addition, the fine sediments injected into the floor of the caves through microcracks and microchannels drainage are formed in an anaerobic environment. Finally, as the dissolution of silica and iron is very slow [61] the iron-developed caves of the IQ have taken a long time to form and have been isolated from the surface [20]. The chemical and physical properties of the caves indicate that the microbiomes inhabiting their interiors have to possess mechanisms to survive in such peculiar environments. The characteristics and aspects of the environment in IQ form this singular environment, which adds even greater scientific interest as it is a key area for the conservation efforts focused on protecting microbiological diversity. These conservation efforts also favor the investigations about bacteria associated with rare plants of ferruginous rocky fields as a reservoir of the microbiota of biotechnological and ecological [62,63] and, in addition, protect the plant biodiversity, which includes hundreds of endemic [64–66] and rare species [65] (Supplementary Figure S1). Even with the imminent risk of loss of these environments, studies involving such cavities are still incipient, and the microbiota associated with this geosystem are understudied.

Such studies can contribute to conservation efforts, as well as broaden our understanding of the rehabilitation capacity of areas degraded by mineral extraction, as demonstrated by a recent work involving the mined areas of the ferruginous fields of the Carajás region, also in Brazil [67].

To our knowledge, this work is the first report of the 16S rRNA gene amplicon sequencing data analyses associated with seven ferruginous caves located in the Brazilian IQ. It is important to highlight that, although no control sample has been incorporated in this work (blank sample), possible contamination from water and other reagents used in sequencing [35], if they have occurred, probably interfered very little and equally in the results obtained, as all samples were processed and sequenced using the same inputs. The results obtained showed that when compared to a quartzite cave, ferruginous caves presented high microbial richness (Figure 2), even though a significant statistical difference was not observed. Microorganisms have a biogeochemical function, including biogenesis processes in the iron caves themselves, active participation in iron oxi-reduction cycles, generation of sulfate and phosphate, and a role in the formation of microchannels and speleothems [68–70]. Energy flux within ferruginous caves is not fully understood. In general, microorganisms in cave systems can play key roles in maintaining subterranean biodiversity, allowing nutrient cycling [71] and potentially the survival of plants established above these cavities, as well as of animals. Although the roles of microorganisms and their ecological interactions need further investigation, the fact that extensive branching root systems are established within the structures of the cavities indicates complex ecological functions (Supplementary Figure S4).

Unlike all other cavities, cave 8 showed signs of human interference, such as traces of fire and litter (PET bottles, cigarette packs and plastic bags). Campfires were found in the vicinity of the cave and also about three meters inside it. A comparison between samples revealed that the microbiota in samples collected closest to the anthropogenic intervention zone in cave 8 differed from the microbiota in areas farther from it. Such a distinct spatial difference was not observed in other caves (where more than one sampling point was tested).

It has been demonstrated that caves accessible to the human visitors and the presence of exogenous organic matter affect diversity and richness when compared to undisturbed areas [72]. In this context, the slow genesis and seclusion of the IQ caves have been essential factors in defining and sustaining their microbial diversity. As observed in cave 8, even minor human interventions cause disturbances, indicating a low resilience of these subterranean systems [17]. This further highlights the mineral extraction activities of the IQ region as an important factor contributing to the loss of potential uncharted genetic heritage, as mining of the ores that constitute the cavities leads to a complete destruction of the caves.

Despite similar geomorphological composition to that of cave 8 (canga ceiling and itabirite floor), much greater richness in terms of observed OTUs was observed in caves 1, 4, 5, and 6. As cave 4 has small dimensions of its conduits, only one ceiling sample was analyzed. It presented a diversity of microorganisms very similar to other caves that had more points sampled. Although different points of cave 5 and 6 were sampled, metabolic diversity predictions were quite homogeneous among the samples investigated, demonstrating that the composition of microbial communities is quite conserved. In other words, the distance of the sample collection point from the entrance of the cave did not affect the microbial diversity, which supports our abovementioned speculation about the effect of anthropogenic interference on the microbiota in cave 8.

Beta-diversity analyses, especially using the weighted Unifrac metric, showed that the majority of floor samples clustered together; ceiling samples on the other hand tended to be more scattered. Other factors, such as altitude or location (within or outside protected areas), did not seem to affect the clustering. In contrast to ferruginous caves, floor and ceiling samples from the quartzite cave were quite similar to each other.

It was observed that the phylum Acidobacteria dominated in ferruginous caves, while the phylum Proteobacteria was the most abundant in other kinds of caves [73–76]. This observation can be explained by differences in geomorphological and geochemical characteristics, genesis and/or ecological interactions [10]. Additionally, Acidobacteria are associated with low pH of the environment, which is in agreement with the reference values (pH below 5.09) observed in ferruginous caves. The most abundant genus in the canga caves was *Blastopirellula*, whereas the genera that present the greatest abundance in the quartzite cave included *Aciditerrimonas*, *Bryobacter*, *Rhodanobacter*, and *Nitrospira*.

From the abundance of identified taxa, it was possible to trace a metabolic profile and to identify traits in the microbiota associated with ferruginous caves that could have use in biotechnologies. Although it is known that only a small proportion of microorganisms can be cultivated [77], thus allowing a targeted analysis of the metabolites produced, it is already demonstrated that it is possible to identify and isolate new biomolecules of wide interest without dependence on culture medium [77,78]. Such techniques could be used in the future to analyze samples from the ferruginous caves, in order to support or refute the conclusions of this study that are described below.

Out of the 328 metabolic pathways identified at level 3, six were selected based on their biological importance and biotechnological interest. Except biosynthesis of 12, 14, and 16 membered macrolides and biosynthesis of type II polyketide products categories, the quartzite cave samples contained reduced proportions of sequences associated with metabolic predictions in all other selected categories.

Macrolides and novobiocin are antibiotics naturally produced by some species belonging to the *Staphylococcus* and *Streptomyces* genera, as well as Cyanobacteria [79–81], and are of wide interest to the pharmaceutical industry [82–84]. However, polyketides are an important and structurally diverse group of secondary prokaryotic metabolites that have several biological functions, but have also been widely reported as molecules with antibiotic and immunosuppressive activities [85]. Considering that the main bacterial genera that produce these compounds belong to the phylum Actinobacteria, which was detected in higher proportions in the caves 8, 7, and 2 (averages 16.6%, 15.4 % and 14.2% of the total OTUs of the floor, ceiling, and biofilm samples from each cave, respectively), the natural cavities could become a promising source of isolates that might find use in the production of new drugs. In addition to potential biotechnological applications, Acidobacteria play important ecological roles, such as plant biomass degradation [86], and plant growth promotion [87].

Adaptive responses to the environment are dependent on signal transduction mechanisms, which can activation secretion systems in prokaryotes [88]. These systems, in turn, can secrete toxins into the environment. Therefore, these categories are related to one another, and may be related to the ecological complexity established in these environments [89].

In caves of other lithologies, the importance of nitrogen fixation and the nitrogen cycle have already been reported as fundamental to maintain the entire ecosystem [90]. As Desai and collaborators (2013) have suggested, diazotrophy might be widespread in chemosynthetic communities in dark caves. Elevated abundance of chemosynthetic organisms (including *Aquisphaera*, *Ferrovum*, *Georgfuchsia*, *Methylocystis*, and *Propionibacterium*) were determined by the metagenomics analysis, indicating the occurrence of fixation of atmospheric nitrogen in the ferruginous caves.

The Chapada de Canga, located in the east of the IQ, extends for about 3900 ha [91] and has a unique biological, archaeological, speleological, geological, and landscape heritage [91]. Despite these characteristics, this region is surrounded by areas of mining and tailing dams, one of which ruptured in 2015, causing irreparable environmental, economic, and social harm [30,92,93]. Three out of the seven caves selected for this study are located in the region affected by the dam's spill. Our work has shown that each of these three caves presented completely different diversity, despite similar morphology, dimensions, and proximity to each other. It was found that, although the richness of cave 3 is the smallest of all analyzed in relation to the abundance of OTUs, its diversity is the largest among

the Chapada de Canga. Another feature of the caves of Chapada de Canga is related to the abundance of specific phyla. The phylum Actinobacteria includes bacteria that are widely studied as they present potential antagonistic activity to human, animal, and plant pathogens [94] which contributes to give importance to the caves of Chapada de Canga, where abundances of this phylum were identified. Bacteria belonging to this phylum have been detected in several environments, including extreme habitats [95–97], and have been characterized as potential producers of biomolecules with anticancer and antimalaria action among other pharmacological interests [98]. These findings are in agreement with the functional metabolic predictions in categories shown, and indicate that cavities could serve as a source of organisms for a new biomolecule production.

A biofilm hanging from the ceiling was identified in cave 2 and sampled for metagenomic analysis. The mineralogy of speleothems is diverse, including, e.g., oxides and hydroxides of iron, phosphates, and sulfates [99], which affect the composition of microbiomes that, on the other hand, play a fundamental role in the mineral deposit formation [100]. The prokaryotic community found in the biofilm was more similar to the floor samples than the ceiling samples, and included genera unique to each of the niches.

This high correspondence between biofilm and floor samples was also seen in functional metabolic predictions. In almost all pathways of secondary compound metabolism and for most of the cell pathways investigated, there was a direct correspondence between the floor and biofilm, except for the nitrotoluene degradation pathways, novobiocin biosynthesis, and bacterial toxins in this context. This result dynamically highlights two important biological events commonly observed in cave environments: the origin of the microbiota associated with the biofilm as a product of the weathering of the soil and the canga above the cavity, and the biofilm as a physical chemistry structure for transferring the associated microbiota to the floor [101].

Results showed that the genera associated with biofilm hanging from the ceiling have a high capacity for oxidation and reduction in iron, phenomena directly associated with the genesis process of the cavities [20,68]. Among the OTUs unique to the biofilm (that were not present in the floor and ceiling samples from the same cave), the most abundant genera were *Ferrovum*, *Geobacter*, and *Sideroxydans*, described as iron oxidizers [102–104]. *Ferrovum* and *Sideroxydans* species are often associated with caves [105], being directly involved with mineral precipitation and thus formation of biofilm [106,107]. In addition, *Geobacter* species have been described as capable of reducing toxic and radioactive metals [108,109], and they therefore play an important role in bioremediation of contaminated sites [110,111].

*Methylocapsa*, *Proprionibacterium*, *Streptacidiphilus*, and *Pseudonocardia* were identified as unique genera in cave 2 ceiling samples. *Methylocapsa* species have already been described as capable of fixing atmospheric nitrogen [112], whereas bacteria belonging to the genera *Proprionibacterium*, *Streptacidiphilus*, and *Pseudonocardia* have been described as species with high potential for antifungal activity [113–116]. Additionally, all these genera are directly or indirectly associated with nutrient cycling (e.g., recalcitrant organic compound and organic matter; Atashgahi et al. [117]) maintenance of biogeochemical cycles (carbon; [118] and nitrogen; [119]), and other metabolic functions (fixation, reduction, and oxidation of metals). These organisms are therefore suitable for several biotechnological applications, such as rhizoremediation [120], effluent treatment [121], promotion of plant growth [120], plant pathogen control [122], and metal biotransformation [123]. In the floor samples, the genera *Burkholderia*, *Rhodanobacter*, and *Variovorax* dominated. Some species of *Burkholderia* are of broad agricultural and industrial interest [124,125]. *Rhodanobacter* has been reported to remove uranium from uranium-contaminated sites [126–128], and *Variovorax* has been described as a genus capable of degrading metaldehyde [129].

## 5. Conclusions

This work presents, for the first time, the microbial diversity associated with ferruginous caves from the Brazilian Iron Quadrangle identified by high-throughput sequencing of the 16S rRNA gene, bringing new findings that could potentially be used in biotech-

nology. In addition, great variations in richness and abundance were observed among ferruginous caves, including those the Chapada de Canga region that are geographically extremely close. Comparisons of prokaryotic diversity in floor and ceiling samples also showed differences in microbiomes within individual caves, which may have a direct correlation, respectively, with the chronology and geomorphological processes linked to the genesis of these cavities. Finally, the data presented in this study highlight the need for conservation of these areas exposed to intense anthropogenic pressure.

**Supplementary Materials:** The following are available online at https://www.mdpi.com/article/10.3390/d13100494/s1, Figure S1: Maps characterizing the iron quadrilateral region, Figure S2: Diversity indices, Figure S3: Functional metabolic prediction, Figure S4: Plant root system throughout the structure of the cavities, Table S1: Sequencing data of collected samples, Table S2: Beta-Diversity indices analysis, Table S3: The most abundant OTUs by sample and the unique OTUs by cave, Table S4: Relative abundance (%) of all OTUs identified, and Table S5: Features of the genera found on the floor, ceiling, and biofilm hanging from the ceiling of cave 2 in Chapada de Canga.

**Author Contributions:** C.G.C.L., M.M.V., L.M.M., L.H.Y.K. and F.F.d.C. collected the cave samples. C.G.C.L., L.O.P., É.B.F. and L.M.M. conceived and designed all experiments. C.G.C.L., L.M.M., L.Â.G.M., S.E.S.G. and J.C.S. analyzed the biological data. L.M.M., L.Â.G.M. and J.C.S. contributed with reagents, materials, and analysis tools. C.G.C.L., L.M.M., P.N.C.V. and A.A.R.S. prepared the figures and tables. C.G.C.L., M.M.V., É.B.F., L.O.P., R.A.B.A., I.F.C., N.P.F., L.C.C.G., W.L.C., L.Â.G.M., F.F.d.C., L.H.Y.K., P.N.C.V., A.A.R.S., J.C.S. and L.M.M. wrote, revised, and approved the final version of the paper. All authors have read and agreed to the published version of the manuscript.

**Funding:** This research was financed in part by the Coordenação de Aperfeiçoamento de Pessoal de Nível Superior—Brazil (CAPES)—Finance Code 001—(the BIGA Project, CFP 51/2013, process 3385/2013), Conselho Nacional de Desenvolvimento Científico e Tecnológico (CNPq Process 481226/2013-3), and Fundação de Amparo à Pesquisa do Estado de Minas Gerais—FAPEMIG (process APQ-02387-14 and process APQ-02357-17). L.M.M., J.C.S., A.A.R.S. have a research fellowship from CNPq. L.M.M. has UFOP grants. The funders had no role in study design, data collection and analysis, decision to publish, or preparation of the manuscript.

**Institutional Review Board Statement:** Not applicable.

**Data Availability Statement:** The original contributions presented in the study are publicly available. This data can be found in the Supplementary Materials and here https://www.ncbi.nlm.nih.gov/sra/?term=PRJNA575049. Registration date: 30 September 2019.

**Acknowledgments:** We thank the members of the Laboratory of Genomics and Bacteria-Environment Interaction of the Federal University of Ouro Preto (UFOP) and members of the Bioinformatics Laboratory of the University of São Paulo (USP) for scientific support, especially Carlos Morais Piroupo by the help with the 16S rRNA gene amplicon sequencing data submission.

**Conflicts of Interest:** The authors declare no conflict of interest.

**Ethics Approval and Consent to Participate:** The cave samples acquisition and analysis were authorized by SISBIO process 54015-1 (authentication code: 13676514).

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
