# Peer review of "16S rRNA Gene Amplicon Sequencing Data of the Iron Quadrangle Ferruginous Caves (Brazil) Shows the Importance of Conserving This Singular and Threatened Geosystem"

_diversity, doi:10.3390/d13100494_

Round 1

Reviewer 1 Report

The weakest point of this study, which the authors mention themselves (lines 460-464), is the absence of  control samples.

Unclear remains the human interference which is mentioned in the MS (lines 476...)

This study is mainly descriptive and not hypothesis driven

Refreshing is the notion of the protection of microbiological diversity (lines 448-458), generally a neglected area in diversity studies.

Interesting is the potential for the discovery of new biomolecules (line584...), I do hope the authors will further explore  this topic and report on it. 

Reviewer 2 Report

General commentary

This manuscript studies the bacterial diversity of ceiling and floor in several ferruginous caves of the Iron Quadrangle in Brazil as compared to the bacterial diversity of a quartzite cave in the same area. This article is interesting and provides original results in terms of biodiversity knowledge.

A large part of my commentaries are more about style than about content, but some issues need to be fixed before the manuscript will be suitable for a publication. Some precisions have to be provided in the Materials & Methods, and in particular in the processing of the data supplied by the Ion Torrent sequencing. R and QIIME2 are powerful tools, but they are not so easy to handle for non-specialist. Thus, the description of the treatment of your data thanks to these tools have to be very precise to avoid any confusion for readers that would like use the same kind of processing.

I also think that the description of your results concerning metabolic analysis needs to be improved.

Please find detailed commentaries following.

Particular commentaries

Introduction

L. 121: “mining industry. Therefore”

Materials and Methods

L. 145-158: I think you should invert the order you provide the information in this paragraph. You begin by the most particular point, how you sampled, to finish with the most general point, how you choose the sampled caves. You should begin with the way you choose the caves, followed by the way you choose the points to sample into the caves and to finish with the way you made the samplings.

L. 148: If I am right, Falcon is a brand owning to Corning Company. In the next paragraph (2.2), you give the name of the product followed by the name of the company and the country in parenthesis, e.g. “NanoDrop (Thermo Fisher Scientific, USA)”: keep the same kind of nomenclature everywhere in the manuscript.

L. 155-158: These temperatures, RH and pH are the ones that you measured in the sampled caves or general data provided by the bibliography, as suggested by the presence of the reference? If you measured these parameters, you should include the data in the table 1.

L. 158: What for is this pH: the floor of the caves, your samples?

L. 161-162: “PowerSoil DNA Mobio kit (GeneWorks, Australia)”: If I am right, this kit is sold by the QIAGEN Company. Maybe GeneWorks is the Company that made extractions for you, isn’t it? If that is the case, you have to clarify it.

L. 175: Country for Invitrogen.

L. 176: Biocycle should be the model of the thermocycler, add the company and the country.

L. 179: Country for GE Healthcare.

L. 187: There is something wrong in the pagination that start again from this page.

L. 194-195: Demultiplexing before the quality control?

L. 205-207: Why did you filtered the reads again if it was already done with the Torrent Suite? If the Torrent Suite process was of too low-quality, I advise you only write about the second filtering to not confuse the reader.

L. 210: “28,252 reads”: how did you set this threshold?

L. 225: “alpha and alpha-phylogenetic plugin”: As provided by QIIME 2 the name of the plugin is “Diversity” and alpha and alpha-phylogenetics are “pipelines” implemented in this plugin. It is important to be precise since QIIME 2 uses the word Plugin.

L. 229-230: You explained just before that QIIME 2 was used to calculate alpha-diversity. Why did you also calculate alpha-diversity with Phyloseq? Similarly, why did you calculate beta-diversity with Phyloseq and not with QIIME 2, which also allows doing it? If this is for a question of graphical representation of the results that is more user-friendly with R than with QIIME 2, I understand, but in this case, it is not necessary to explain that you first calculated with QIIME 2.

L. 240: “Bray-Curtis”

L. 242 and L. 244-245:”beta and beta-phylogenetic plugin” and “beta-group-significance plugin with pairwise comparison”: it sounds like functions of the Diversity plugin implemented in QIIME 2 and not like Phyloseq package functions. Did you used QIIME 2 or Phyloseq? To do what with each one?

L. 245: I guess you did not include the Biofilm sample because having only one repetition is not statistically relevant, but you should explain it.

L. 258: Remove the point after “software”.

Results

L. 264-266: This information should not take place in the results: it is your material at the beginning of the study. You should move this sentence in 2.1 because it is an interesting summary of table 1.

L. 279: What about cave 3, which seems to be the one with the lowest number of OTUs?

L. 281:”there was no statistical difference”: is this statement about the 8 caves all together or only about Chapada caves, which you speak about in the preceding sentence?

In addition, where is the table depicting the statistical results concerning the comparison between caves regardless of coming from ceiling or floor? Figure S2 is only about the comparison ceiling vs. floor, isn’t it? Or else the last line of Figure S2 (b) gives this result? This would be confusing since you depict the result given in the table as “ceiling vs. floor”?

L. 292-294: This sentence comes under the discussion, move it.

L. 297-300: Same commentary, it comes under the discussion, especially since it obliges you to refer to figure S3 while you don’t have given yet the results about function metabolism prediction.

Figure 2: In (a), you added an * to show the control cave. You have to explain it in the figure caption. You could also add an * to cave 7 in (b) to provide consistent information.

In (b), it is a shame not keeping the same color associated to each cave than in (a).

L. 310-312: Discussion.

L. 306: “soil prokaryotic”: remove “soil”: it is confusing as you also give the results about ceilings.

Figure 2 + Table S2: In Figure 2, you only present the graphical results about weighted and unweighted unifrac. I do not think that it necessary to present the results for Aitchison, Bray-Curtis and Euclidian in Table S2 and neither to write that you performed these analyses in the M&M. All these 5 methods calculate the same thing: a distance matrix. There are small differences between these calculation methods, but you do not explain what these differences are and there is no difference in your result according to these methods. Thus, I advise you to simplify your manuscript by omitting the methods that you do not really develop.

L. 315: In the preceding paragraph, you depicted statistical results for beta-diversity ceiling vs. floor, where are the statistical results cave vs. cave?

L. 342-345: You wrote the same explanation two times: remove the first sentence or remove the second and the third sentence.

In addition: Is this greater abundance statistically relevant or is it only a trend? Is it a statistical test associated to this statement?

Paragraph 3.5 + Figure 5: Please better explain how you choose the subcategories to highlight. Figure 5(a) does not reveal particular differences between ferruginous and quartzite cave microbiota regarding the highlighted subcategories to me.

In figure 5(b), how can you write that there is a low diversity in cave 7 regarding the highlighted pathways while there is a great diversity in cave 2 and 8? Figure 5(b) shows proportions and not diversity, isn’t it? The size of the boxplot indicates a great variability between samples in caves 2 and 8 and a low variability between samples in cave 7, but it is not in relation with the diversity in each cave? L. 367-368, you explained that an anova test was performed: how are represented the results of these anova on the figure? I certainly not properly understand the figure, but the figure is also not properly explained.

L. 373-375: Differences that are not statistically significant.

L. 376:”the highest richness”: replace “richness” by “diversity”.

L. 380:”the highest richness”: replace “richness” by “diversity”.

L. 383-385: This sentence comes under the discussion, move it.

L. 397:”which is a rare geoform in ferruginous caves [22]”: this sentence comes under discussion or introduction. No reference is needed in the results. Please, move this sentence.

L. 415-416: How are represented the results of these anova on the figure?

L. 423-435:”phenomena…” to the end: discussion.

Discussion

L. 436: Skip a line.

L. 445-446: “needed to develop mechanisms”: be careful with anthropomorphism, this sentence suggest an intention. Maybe “have to possess mechanisms”, or something close, would be better.

L. 465-466: This statement is not statistically significant, be most careful in wording.

In addition, no return to the figures in the discussion.

L. 475: Ok, in that case the return to the figure is tolerable because it is a figure that you did not present before and that it not really comes under results.

L. 478:”inside the.”: replace “the” by “it”

L. 480-481: Remove “(Supplementary Figure 3)”.

L. 493:”much greater highest”: remove “highest”.

L. 501: Remove “(Supplementary Figure 3)”.

L. 504-505: Explain what you mean.

L. 509-510: Remove “(Figure 3 and 4)”.

L. 515: “genera_that”.

L. 519: Remove “(Figure 5A)”.

L. 525-526: For their biological importance and not because of significant differences between ferruginous and quartzite caves. You have to explain this choice in the results.

L. 526-527: I definitely do not understand the results about metabolism: the figure 5(b) displays the lowest proportion of sequences associated to nitrogen metabolism in quartzite cave to me, the size of the boxplot being the result of the heterogeneity between samples coming from a same cave.

L. 546-554: You wrote the opposite at L.526-527.

L. 559: Remove “Supplementary Figure 5”.

L. 569: Remove “(Figure 6A)”.

L. 572: Remove “(Figure 2A)”.

L. 573-574: “the richness of cave 3 is the smallest”: in terms of OTUs abundance. I think you should better to write about OTUs abundance than richness that is more confusing.

L. 579: Remove “(Figure 6BCD)”.

L. 580: “have been detected in extreme habitats.[113-115]”: yes, but not only, reformulate the sentence.

In addition, remove the unnecessary point.

L. 583: Remove “shown in Figure 5B”.

L. 590: Remove “(Figure 7A)”.

L. 592: Remove “(Figure 7B)”.

L. 595-596: ”This result… commonly observed in cave environments”: references?

L. 600-602: Why do you put the references [119-12] here? This sentence is about your results and not about bibliography. Or else, the reference justify that these genera are known to be iron oxidizers. In that case, the sentence needs to be reformulated.

L. 614: A second closing parenthesis is needed.

L. 619: “[139])”: remove the unnecessary parenthesis.

L. 622-629: It rather comes under conclusions, move this paragraph.

Conclusions

L. 630: Skip a line.

Reviewer 3 Report

Some issues/questions about the paper are given in the first part of the review. The second part contains edits and minor suggestions. Overall excellent work. It is obvious you worked hard on this study and the paper.
